# Interleukin-13 rs1800925/-1112C/T promoter single nucleotide polymorphism variant linked to anti-schistosomiasis in adult males in Murehwa District, Zimbabwe

Emilia T. Choto[1]*, Takafira Mduluza[2],[¤], Moses J. Chimbari[1]

1 School of Public Health Medicine, College of Health Sciences, University of KwaZulu-Natal, Durban, South Africa, 2 School of Laboratory Medicine and Medical Sciences, University of KwaZulu-Natal, Durban, South Africa

☯ These authors contributed equally to this work.
¤ Current address: Biochemistry Department, University of Zimbabwe, Harare, Zimbabwe
* emiliachoto@gmail.com

**Data Availability Statement:** All relevant data are within the manuscript and its Supporting Information files.

## Abstract

### Background

Chronic schistosomiasis is predominantly induced through up-regulation of inflammatory cytokines such as interleukin (IL)-13. IL-13 may contribute to the disease outcomes by increasing eosinophil infiltration thereby promoting fibrosis. IL-13 may act as an immuno-suppressive inflammatory cytokine that may promote carcinogenesis and also may offer protection against schistosomiasis thereby reducing risk of schistosome infections. Our study evaluated the frequency of the IL-13 rs1800925/-1112 C/ T promoter single nucleotide polymorphisms (SNPs) among schistosomiasis infected individuals and assessed the association of the variants on IL-13 cytokine levels. We also investigated IL-13 rs1800925 polymorphisms on prostate-specific antigen levels as an indicator for risk of prostate cancer development.

### Methodology

The study was cross-sectional and included 50 schistosomiasis infected and 316 uninfected male participants residing in Murehwa District, Zimbabwe. IL-13 rs1800925 SNPs were genotyped by allele amplification refractory mutation system-polymerase chain reaction. Concentrations of serum prostate-specific antigens and plasma IL-13 were measured using enzyme-linked immunosorbent assay.

### Results

Frequencies of the genotypes CC, CT and TT, were 20%, 58% and 22% in schistosomiasis infected, and 18.3%, 62.1% and 19.6% in uninfected participants with no statistical differences. There were significantly (p<0.05) higher IL-13 cytokine levels among both infected and uninfected participants with the genotypes CC and CT; median 92.25 pg/mL and 106.5

**Funding:** The research study was supported by the OAK Foundation and the University of KwaZulu-Natal - College of Health Sciences Scholarship for operational funds (identifier number 636753). All authors are supported by OAK Foundation. The funders had no role in study design, data collection and analysis, decision to publish, or preparation of the manuscript.

**Competing interests:** The authors have declared that no competing interests exist.

pg/mL, respectively, compared to TT variant individuals; 44.78 pg/mL. Within the schistosomiasis uninfected group, CC and CT variants had significantly (p<0.05) higher IL-13 levels; median 135.0 pg/mL and 113.6 pg/mL, respectively compared to TT variant individuals; 47.15 pg/mL. Within the schistosomiasis infected group, CC, CT and TT variant individuals had insignificant differences of IL-13 level. Using logistic regression, no association was observed between prostate-specific antigen levels, IL-13 cytokine levels and IL-13 rs1800925 variants (p>0.05).

## Conclusion

IL-13 rs1800925 C variant individuals had the highest IL-13 cytokine levels among the schistosomiasis uninfected suggesting that they may be protective against *Schistosoma* infections. There was no association between IL-13 concentrations or IL-13 rs1800925 variants and risk of prostate cancer indicating that IL-13 levels and IL-13 rs10800925 may not be utilised as biomarker for risk of prostate cancer in schistosome infections.

## Introduction

Schistosomiasis is a neglected tropical parasitic disease caused by the digenetic trematodes of the *Schistosoma* genus [1,2]. The disease contributes to a globally estimated 70 million disability-adjusted life years (DALYs) [3] and accounts for 200 000 deaths annually in the sub-Saharan region [4]. Factors that contribute to the infection include poor sanitation, and open water bodies infested with schistosome specific vectors and lack of knowledge about the disease [5]. Cardiopulmonary, gastrointestinal tract, genitourinary tract and the central nervous system are the most affected human functional systems by acute and chronic schistosomiasis infections [5].

Acute schistosomiasis infections are a result of T helper (Th) 1 immune response whilst chronic infections are a result of Th2 immune response induced through production of the cytokines such as Interleukin (IL)-13 [6,7]. IL-13 is a small molecular regulatory multifunctional cytokine mostly produced by activated lymphocytes that modulate inflammatory processes [8]. IL-13 promotes Immunoglobulin (Ig) E production and class switch, up-regulates major histocompatibility complex (MHC) class expression II and it inhibits inflammatory Th1 cytokine production [8–10]. Schistosome infection associated cytokines are connected to cytokine gene polymorphisms of individual variability that influence immune responses and disease outcomes [11]. Cytokine gene polymorphisms such as single nucleotide polymorphisms located in the promoter region of encoding genes may modify gene transcription and cytokine production in parasitic infections and autoimmune diseases [11]. IL-13 plays a major role in egg granuloma formation during schistosome infections and can stimulate fibroblasts to make collagen leading to fibrosis that may lead to portal hypertension, thus causing much of the morbidity and mortality associated with schistosomiasis [12–14]. High IL-13 is associated with the development of hepatic fibrosis [15,16] and periportal fibrosis [17].

Through several mechanisms, IL-13 promotes and combats cancer progression [18,19]. IL-13 plays a critical role in down-regulation of tumour immunosurveillance and is central to a novel immunoregulatory pathway in which natural killer T cells are induced by tumours to secrete IL-13, which acts through intermediate cells to suppress cytotoxic T cell responses against the tumour [20]. IL-13 can promote survival of certain types of tumours through direct

action on the tumour or acting through suppression of immuno-surveillance [21]. IL-13 stimulates changes in epithelial and smooth muscle cell functions leading to hypersensitivity reactions [22–24]. Epithelial lesions, tumours, and ulcers have been associated with the presence of *S. haematobium* eggs in the lower genital tract [25]. High levels of Th2 cytokines including IL-13 have been observed in the tumour microenvironment and peripheral blood of individuals with bladder, breast and prostate cancers [26,27].

Single nucleotide polymorphisms (SNPs) are the most common genetic variations and if located in the coding regions of genes can result in loss, abrogation or altered function of the downstream protein by causing alterations in amino acid sequences and protein structure [28,29]. IL-13 gene is located on chromosome 5q23.31 and encodes IL-13 cytokine [30]. Single nucleotide polymorphisms substitution of the nucleotide cytosine (allele C) to thymine (allele T) at the IL-13 rs1800925 (also known as -1111C/T, -1055C/T and -1112C/T) site in the promoter region, results in the binding of nuclear proteins [31]. This causes overproduction of IL-13 cytokine in Th2 lymphocytes that may play a role in allergic and chronic inflammatory diseases [31–33]. Isnard *et al.* (2011) showed that *S. haematobium* infection levels are associated with interleukin-13 (rs7719175) gene promoter polymorphism [34]. In 2012, Gaitlin *et al.* showed that protection against severe infection with *S. mansoni* was driven by functional IL-13–1055 polymorphisms [35]. IL-13 rs1800925T has been associated with a higher risk of pathological hepatic fibrosis in *S. japonicum* infected individuals [36]. Additionally, Kouriba *et al.* showed that IL-13 -1112T variant was associated with susceptibility to *S. haematobium* infection [37]. In contrast to the above findings, recently Adedokun *et al.* (2018) found no statistical difference in the IL-13 rs7719175 genotypic or allelic frequencies between schistosome-infected and uninfected controls or any association with disease [38].

Limited reports on *Schistosoma* ova and prostate cancer co-existence have been reported suggesting possible prostate cancer evolution due to schistosomiasis infection [39–42]. Prostate cancer is one of the most common cancer in Zimbabwe [43]. Recently, Choto *et al.* (2020) [44] showed that *S. haematobium* egg burden was associated with the risk of prostate cancer development in adult males [45]. Schistosome induced pro-inflammatory and immunosuppressive cytokines have been reported to arbitrate intracellular communication, regulate gene transcription and promote carcinogenesis in different tumour types including prostate cancer [45–47]. Screening for prostate cancer is initially done by using prostate-specific antigen levels to detect the diseases early stage for better management and reduction of disease specific mortality [48]. Whilst prostate-specific antigen levels above 4.0 ng/mL serves as a reference point for further prostate management [49]. To further determine risk of prostate cancer, combination of prostate-specific antigen with single nucleotide polymorphisms was shown to be effective in men with prostate-specific antigen levels greater than 4 ng/mL [50]. Combination of genetics and the PSA test is useful for predicting the risk of prostate cancer that enables stratifying the population into different risk groups that may be a basis for the development of personalized screening for prostate cancer [50,51].

Characterizing cytokine genetic variability on the association of schistosome-induced cytokines is important in understanding disease burden and other disease outcomes such as prostate cancer in schistosome endemic countries. Therefore, the aim of the study was to evaluate the frequency of the IL-13 rs1800925/-1112 C/ T promoter gene polymorphisms. We assessed the association of the IL-13 rs1800925/-1112 C/ T promoter variants on IL-13 cytokine levels between schistosome infected and uninfected male individuals. We investigated IL-13 rs1800925/-1112 C/T promoter single nucleotide polymorphisms and susceptibility to schistosomiasis. Furthermore, we investigated the implications of IL-13 rs1800925/-1112C/T variants on the risk of prostate cancer development in male individuals residing in Murehwa District, Zimbabwe.

## Materials and methods

### Study design, study area, study population

This is a sub-study of November 2019 prostate cancer and schistosomiasis cross-sectional study that included 366 male adult participants recruited from Murehwa District, Mashonaland East province in Zimbabwe where *Schistosoma haematobium* is endemic [44]. The study area was chosen because it has a high schistosomiasis burden of 47.4% in school aged children a representative of the study population [52]. Overall population of Murehwa District is 199 607 and about 94 000 individuals are men [53]. The District consists of more than 90% rural areas where majority of the residents rely on nearby rivers for their domestic needs and farming activities. Sanitation in the area is poor with the majority of residents relying on unsafe drinking water and open defecation. The main study assessed possibility of prostate cancer development due to schistosome infections hence, only male adult individuals were enrolled. The inclusion criteria for the study participants were: adult male individuals aged 18 years and above, with and without schistosomiasis (*S. mansoni* and *S. haematobium)* who are residents of the study area that gave consent to be part of the study. The exclusion criteria included being under 18 years of age and those not consenting to participate. Adults aged 18 years and above were included into the study because they were able to understand the study and voluntarily gave informed consent.

### Ethical approval

Ethical approvals were obtained from the University of KwaZulu-Natal, Biomedical Research Ethical Committee (UKZN-BREC BF689/18) and the Medical Research Council of Zimbabwe (MRCZ/A/2128). Consent to conduct the study in Murehwa District was obtained firstly from the community gatekeepers. Prior to enrolment, the study aims and procedures were explained to all participants in the local language (Shona). Written informed consent was obtained from participants. Participants were free to withdraw from the study without prejudice to any services offered at the study sites.

### Sample size calculation and sampling

A sample size of 245 including 20% added on to account for drop out was calculated [54] based on the inferred schistosomiasis prevalence for adults of 15.8% (a third of the 47.4% for the school aged children) [52]. We purposely recruited male adults aged 18 years and above residence of Murehwa District, Zimbabwe. However, the number of participants recruited ended up being 366 because more men met the study criteria [44]. Hence, the sample can be considered representative of a larger population in the study area.

Participants from rural villages in Murehwa Disctrict were invited to be part of the study through invitation by the village health workers to report to 8 sampling centres that are used by the community for meetings or health education and immunisation programs (such centres include schools or primary health centres) for enrolment into the study. The sampling centres were Jekwa rural clinic, Dombwe rural clinic, Mutize primary school, Kareza gathering point, Kapasura gathering point, Magaya primary school, Guzha primary school and Inyagui primary school. Participants enrolled into the study provided their age or date of birth, samples for parasitological diagnosis and blood sample for serological assays.

### Parasitological diagnosis

Urine samples collected between 10:00 hours and 14:00 hours were processed within 2 hours of sample collection. The urine was processed and examined to diagnose *S. haematobium* infections as described by Mott *et al.* 1982 [55]. Urine sample collection and processing procedure was

repeated on three consecutive days. The number of eggs were expressed per 10 mL of urine. Stool samples were collected, processed, and examined using the Kato–Katz method [56]. The *S. mansoni* eggs were expressed per milligram of stool. Participants were diagnosed positive for schistosomiasis infection if any of the two species of the parasite egg was detected in their urine or stool samples. *S. haematobium* infection intensity was classified in accordance with the World Health Organization guidelines; no infections 0 eggs, light infection <50 eggs/10mL and heavy infection >50 eggs/10mL [57]. *S. mansoni* infection intensity was classified with WHO guidelines; no infections 0 eggs, light infections < 100 eggs per gram (epg), moderate infections ≥100 < 400 epg and heavy infections ≥ 400 epg [57]. Schistosome infected individuals were treated with praziquantel (PZQ) at the standard single oral dose of 40mg/kg per body weight [58].

## Determination of IL-13 concentrations

Serum was obtained from the venous blood into blood collecting tubes without anti-coagulant (BD vacutainers Lot- 7114655). Qualitative detection of IL-13 concentrations were measured by enzyme linked immuno-sorbent assay (ELISA) using human IL-13 development ELISA kit (product number-3471-1H-6; Mabtech Company, Sweden) according to the manufactures intstructions. All samples and standards were measured in duplicate and concentrations were determined from a standard curve using mean optical density values.

## Determination of prostate-specific antigen concentrations

As reported by Choto et al. (2020) [44], qualitative detection of prostate-specific antigen levels was done using enzyme linked immuno-sorbent assay (ELISA) using R &D Human Kallikrein 3/prostate-specific antigen Duo Set ELISA; DY1344 and R&D systems catalog # DY008 ancillary kit (96 well microplates, plate sealers, substrate solution, stop solution, plate coating buffer (PBS), wash buffer, and reagent diluent concentrate) according to the manufacturer's instruction. All samples and standards were measured in duplicate and concentrations were determined from a standard curve using mean optical density values. The sensitivity of a prostate-specific antigen levels above 4.0 ng/mL for detecting prostate cancer range from 63% to 83% hence served as a reference point for further prostate cancer analysis [49,59]. Serum prostate-specific antigen concentrations were expressed as ng/mL and were categorised into two groups according to prostate-specific antigen greater than (>) 4 ng/mL and less than (<) 4 ng/mL.

## DNA extraction protocol

Genomic DNA was isolated from whole blood using the Zymo Quick-DNA™ Miniprep Plus Kit (Zymo Research, Irvine, CA), according to manufacturer's instructions. In summary, whole blood sample was mixed with genomic lysis buffer followed by 10-minute room temperature incubation. Samples were vortexed and incubated in a water bath at 55˚C for cell lysis. An equal volume of genomic binding buffer was added to the digested sample. The mixture was transferred to a Zymo Spin column in a collection tube and centrifuged. DNA pre-wash buffer was added and centrifuged then g-DNA wash buffer was added and centrifuged. The spin column was transferred to an Eppendorf tube and DNA elution buffer was added. DNA elution buffer was added to the spin column, incubated at room temperature and was centrifuged. The DNA was collected in the Eppendorf tube and stored at -80˚C until further analysis were conducted.

## Genotyping

Genotypes for the IL-13 −1112C/T (rs1800925), polymorphism were determined by allele-specific ARMS–PCR methodology as described by Hummelshoy *et al.* [60]. For each DNA

sample, two reactions were done with each of the forward primers: IL-13–1112 forward C primer: 5'-TTCTGGAGGACTTCTAGGAAAAC-3' or IL-13–1112 forward T primer: 5'-TTCTGGAGGACTTCTAGGAAAAT-3'. The reverse primer IL-13 -740R: 5'-GGAGATGG GGTCTCACTATG-3' was used for all samples. Amplicon size of 319 bp product is detected. The specific primer concentrations were 0.5 um for both reactions. Internal control PCR primers (amplicon size of 726 bp) were included in each reaction and allele specific and internal control primer sequences and PCR product sizes as follows: Forward 5'-TGCCAAGTGGAG CACCCAA -3' and reverse: 5'- GCATCTTGCTCTGTGCAGAT-3'. Each PCR reaction was performed under the following conditions: genomic DNA, Taq master mix (Inqaba biotechnical industries, SA) containing 20 mM Tris-HCL pH 25˚C, 1.8 mM magnesium chloride, 22 mM $NH_4Cl$, 22 mM potassium chloride, 0.2 mM deoxynucleotide triphosphates (dNTPs), 25 units/mL one Taq DNA polymerase, 5% glycerol, 0.06% IGEPAL CA-630 and 0.05% Tween-20. The cycling conditions were 2 min at 94˚C, 15 cycles of 30 sec at 94˚C, 60 sec at 63˚C and 60 sec at 72˚C, 20 cycles of 30 sec at 94˚C, 60 sec at 60˚C and 60 sec at 72˚C, and finally 5 min at 72˚C. The PCR products were separated on 2% agarose gel.

## Statistical analyses

Data was analysed using Statistical Package for Social Sciences (SPSS) statistics version 16 and graph pad prism version 6.0. Continuous variables were summarized by median and inter-quartile range (IQR), and categorical variables were summarized by frequency and percentages (%). Genotype frequencies were tested for agreement with Hardy–Weinberg equilibrium using sing the chi-square ($X^2$) goodness of-fit-test analysis based on likelihood theory, using estimates of African genotype rs1800925/ -1112 CC, CT and TT frequencies of 0.340, 0.472, 0.188 respectively available at http://www.ensembl.org/Homo_sapiens/Variation/Population? db=core;r=5:132656617-132657617;v=rs1800925;vdb=variation;vf=50310967. Differences in genotypic and allelic frequencies between schistosomiasis infected and controls were assessed by Chi-square test and strength of association was assessed by odds ratio with a 95% confidence interval. Differences of genotypic variants and different villages were assessed by Chi-square test. Additionally, genotypic differences on all 50 schistosomiasis infected and age matched 50 schistosomiasis uninfected were assessed using the Chi-square test. Descriptive statistics were applied on the following variables; schistosomiasis prevalence and infection intensity. Kruskall-Wallis test and Mann-Whitney test were used to determine the IL-13 cytokine level differences of the following: schistosomiasis infected and uninfected groups, schistosomiasis infection intensity groups, IL-13–1112 CC, CT and TT genotypes and prostate-specific antigen level differences > or < 4 ng/mL. Multiple regression analysis was run to associate the dependent variable; IL-13 levels from IL-13 SNP genotype (CC, CT, TT) and schistosomiasis status. Odd ratios (OR) and their 95% confidence intervals were used to measure the strength of association. Binary logistic regression was done to predicted risk of prostate cancer development using the dependent variable; prostate-specific antigen levels groups (> 4 ng/mL or < 4ng/mL) and independent variables; IL-13 concentrations and schistosomiasis status.

## Results

### Prevalence of schistosomiasis and distribution of IL-13 – 1112C/T genotype

A total of 366 adult male participants were recruited from eight rural schistosomiasis endemic communities within Murehwa District, Zimbabwe, namely Dombwe, Mutize, Kareza, Kapasura, Inyagui, Magaya, Jekwa and Guzha. Participants ranged from 18 to 95 years with a

median age of 44 (31–61). Schistosomiasis prevalence was 13.7% (n = 50) with *S. haematobium* infections prevalence of 12.3% (n = 45) and *S. mansoni* infection prevalence of 1.4% (n = 5). Majority of the infected participants were young adults and they harboured mostly light infections as shown in the participants demographic Table 1. The overall schistosomiasis for both *S. haematobium* and *S. mansoni* prevalence in each village was as follows: Dombwe 3.2%, Mutize 0%, Jekwa 3.5%, Guzha 10%, Magaya 13.2%, Inyagui 11.9%, Kapasura 37.7% and Kareza 13.5%. The above presented data has been previously reported by Choto et al. (2020) [44]. There was unsuccessful genotyping of 10 samples hence, the genotypic frequency distribution of IL-13 -1112C/T polymorphism for 356 participants are shown in Table 2. The frequencies of the genotypes CC, CT and TT, were 20%, 58% and 22% in schistosomiasis infected participants, and 18.3%, 62.1% and 19.6% in participants without schistosome infection, respectively. All genotype frequencies were distributed in accordance with Hardy–Weinberg equilibrium (at $X^2$ (2) = 13.398; P < 0.001) shown in S1 Table. We found no significant differences in the genotypic or allelic frequencies of IL-13 (Table 2) promoter gene polymorphisms between infected and uninfected groups (p > 0.05). Also, there was no significant differences in the genotypic of IL-13–1055 C/T promoter gene polymorphisms and the different villages (p = 0.096, $X^2$ = 21.207, df = 14). Furthermore, there was no significant association between the IL-13-1112 C/T genotype age matched schistosomiasis infected and uninfected (CC 14%; CT 33% and TT 10%) (p = 0.659, $X^2$ = 0.835, df = 2). While slightly higher *S. haematobium* mean egg counts were observed among participants in different villages with the heterozygous genotype CT and homozygous TT variants there were no significant differences among participants from Guzha, Magaya and Kapasura villages (Fig 1).

## IL-13 cytokine levels

IL-13 cytokine concentrations measured were detected in 107 participants and the concentrations ranged from 2.055 pg/mL to 1 334 pg/mL with a median of 85.13 (IQR 37.66–206.20) pg/mL. Two hundred and fifty-nine (259) samples had undetectable IL-13 concentrations. As shown in Fig 2, participants with schistosomiasis (n = 20) had lower but not significant levels of IL-13, 75.64 (14.52–287.50) pg/mL compared to schistosomiasis uninfected group (n = 87) 89.88 (40.03–206.20) pg/mL; p = 0.481. There were significantly higher levels of IL-13 cytokine among participants with the genotypes CC; 92.25 (41.22–210.9) pg/mL and CT; 106.5 (53.09–254.8) pg/mL compared to TT variant individuals, 44.78 (19.86–98.18) pg/mL (p = 0.0163) (Fig 3). Furthermore, there were significantly higher (p = 0.004) IL-13 cytokine levels of individuals with the CT genotype compared to the TT genotype. There was no significant higher (p = 0.086) IL-13 cytokine levels of individuals with the CC genotype compared to the TT genotype as shown in Fig 3. As illustrated in Fig 4, IL-13 cytokine levels of the IL-13-1112 CC and CT variants were not significantly different (p = 0.310) from those with the TT variant for

**Table 1. Participant demographics by age, schistosomiasis status and infection intensity.**

| | Age groups | | | | | | | | | |
| | Young adults | | | Middle aged | | Older aged | | | Missing | Total |
| | 18–19 | 20–29 | 30–39 | 40–49 | 50–59 | 60–69 | 70–79 | 80–100 | | |
|---|---|---|---|---|---|---|---|---|---|---|
| **Schistosomiasis negative** | 10 (3.2) | 45 (14.2) | 63 (19.9) | 58 (18.4) | 41 (13.0) | 56 (17.7) | 20 (6.3) | 19 (6.0) | 4 (1.3) | **316 (100)** |
| *S. haematobium* infected | 8 (17.8) | 11 (24.4) | 11 (24.4) | 6 (13.3) | 2 (4.4) | 5 (11.1) | 0 (0) | 1(2.2) | 1 (0) | **45 (100)** |
| *S. mansoni* infected | 0(0) | 2(40) | 0 (0) | 2 (40) | 0 (0) | 0 (0) | 1(20) | 0 (0) | 0 (0) | **5 (100)** |
| **Light infection intensity** | 5 (11.4) | 13 (29.5) | 11 (25.0) | 8 (18.2) | 2 (4.5) | 3 (6.8) | 1(2.3) | 1(2.3) | 0 (0) | **44 (100)** |
| **Heavy infection intensity** | 3 (50.0) | 0 (0) | 0(0) | 0 (0) | 0 (0) | 2 (33.3) | 0 (0) | 0 (0) | 1 (16.7) | **6 (100)** |

**Table 2. Distribution and association of IL-13-1112C/T genotype and allelic frequencies in schistosomiasis infected and uninfected participants.**

| Genotype/Allele | Schistosomiasis infected n (%) | Schistosomiasis uninfected n (%) | Total population n (%) | p value | $X^2$ | df | Odds Ratio (95% C. I) |
|---|---|---|---|---|---|---|---|
| CC | 10 (20.0) | 56 (18.3) | 66 (18.5) | 0.864 | 0.304 | 2 | - |
| CT | 29 (58.0) | 190 (62.1) | 219 (62.0) | | | | |
| TT | 11 (22.0) | 60 (19.6) | 71 (19.5) | | | | |
| C | 24 (48) | 151 (49.3) | 175 (49.2) | 1 | 0.003 | 1 | 1.014 (0.623–1.649) |
| T | 26 (52) | 155 (50.7) | 181 (50.8) | | | | 0.987 (0.633–1.540) |

schistosomiasis infected. IL-13 cytokine levels for IL-13-1112 CC, CT and TT variants were 89.88 (45.97–326.0) pg/mL, 76.82 (26.98–322.5) pg/mL) and 13.92 (7.989–195.5) pg/mL in schistosomiasis infected participants, and 135.0 (56.65–215.7) pg/mL, 113.6 (59.02–253.7) pg/mL and 47.15 (28.16–9.18) pg/mL for participants without schistosomiasis. Levels of IL-13 were significantly different between the IL-13–1112 CC, CT and TT genotypes of the schistosomiasis uninfected individuals (p = 0.022).

**Association of IL-13 cytokine levels, IL-13 – 1112C/T genotypes, schistosomiasis and infection intensities.** Multiple regression analysis run to determine association of IL-13 levels from IL-13-1112 variants (CC, CT, TT) and schistosomiasis status confirmed no association [$F_{(2, 107)}$ = 0.920, P (0.406) > 0.05, $R^2$ = 0.017]. Despite higher infection intensity of the IL-13 -1112CT, infection intensity of IL-13–1112 CC, TT and CT genotypes were not significantly different as shown in Fig 5. Similarly, IL-13–1112 CT and CC variants had insignificantly higher schistosomiasis burden (mean egg count: 11 (6.0–36.25) eggs/10 mL and 11 (7.0–34.5) eggs/10mL, respectively) compared to TT variants (mean egg count, 9 (4.0–14.0) eggs/10 mL); p = 0.499.

**Association between risk of prostate cancer development and IL-13 (rs1800925) variants, schistosomiasis status; and IL-13 cytokine levels.** One hundred and ninety-five

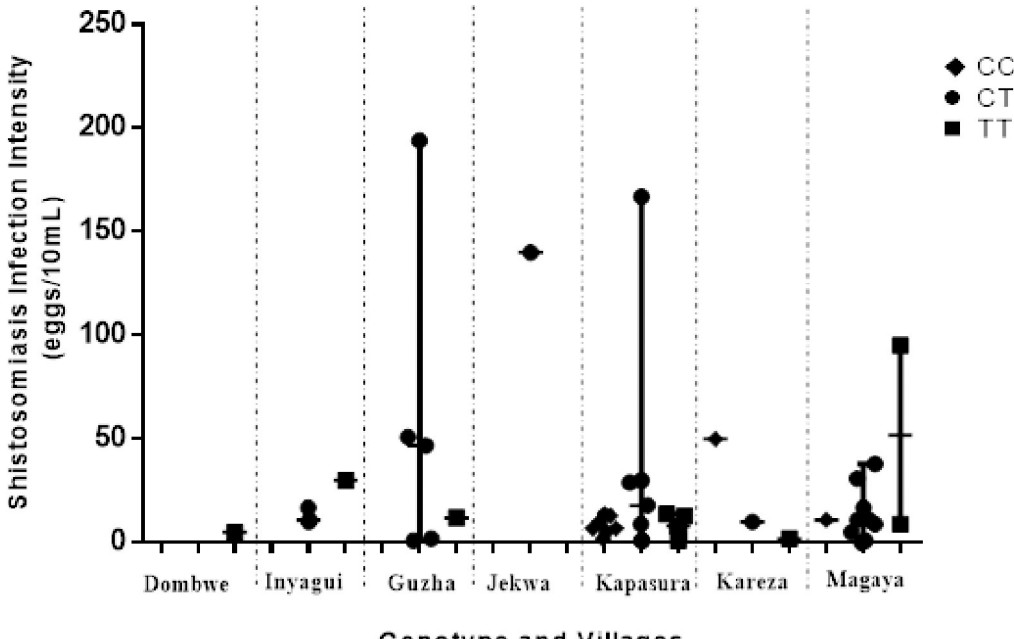

**Fig 1. Comparison of IL-13-1112C/T genotype of *S. haematobium* among individuals in different villages.**

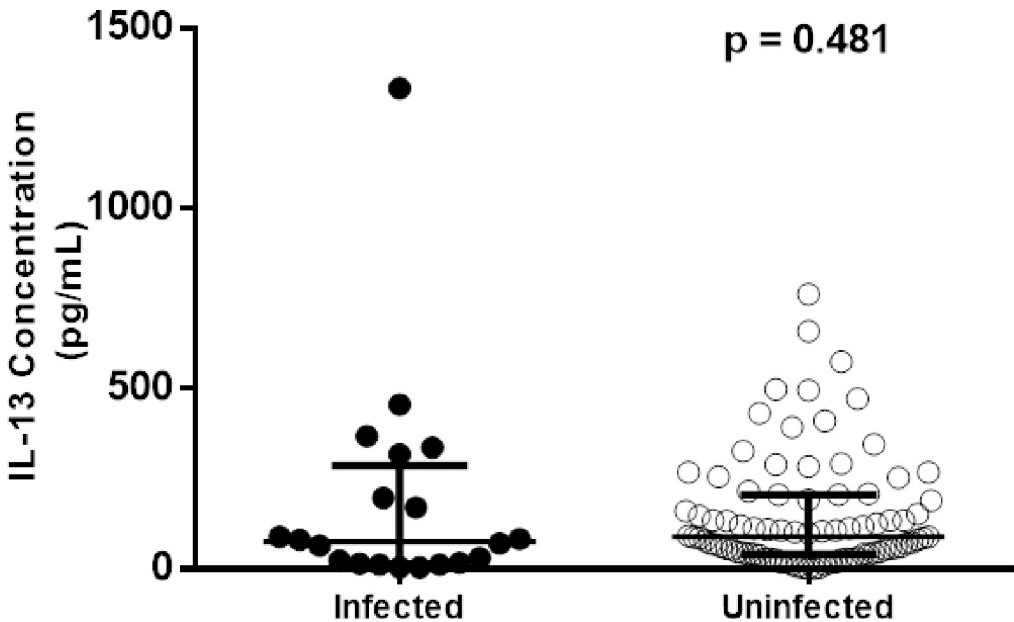

**Fig 2. Comparison of IL-13 cytokine levels in schistosomiasis infected and uninfected.**

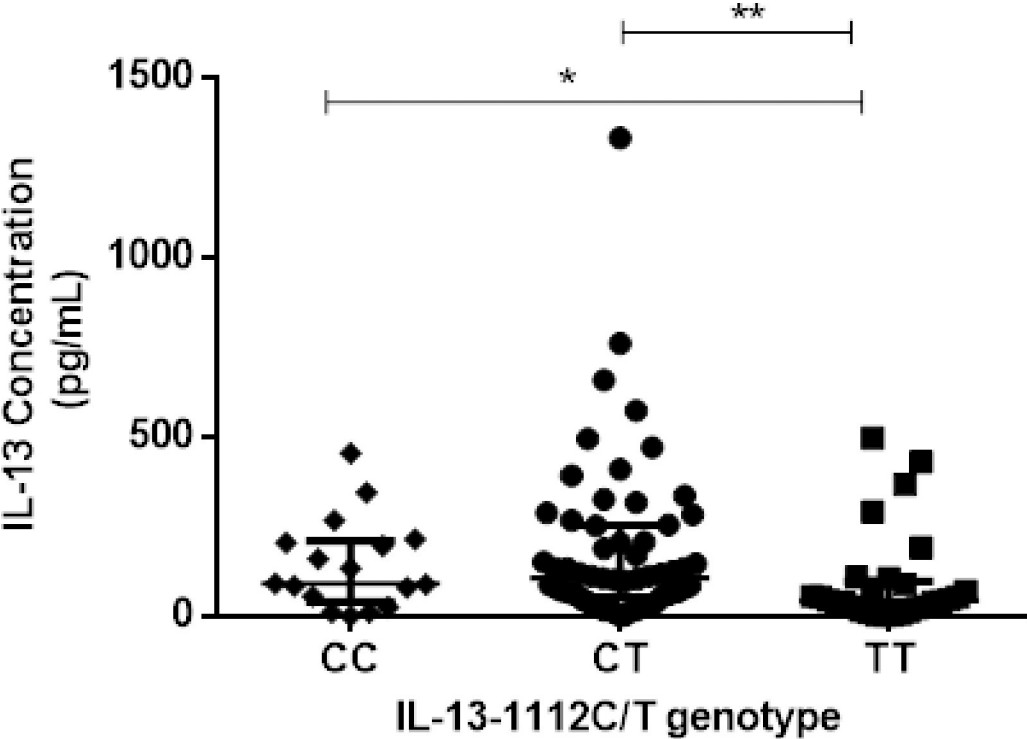

**Fig 3. Comparison of IL-13 cytokine levels in IL-13–1112 C/T variants.**

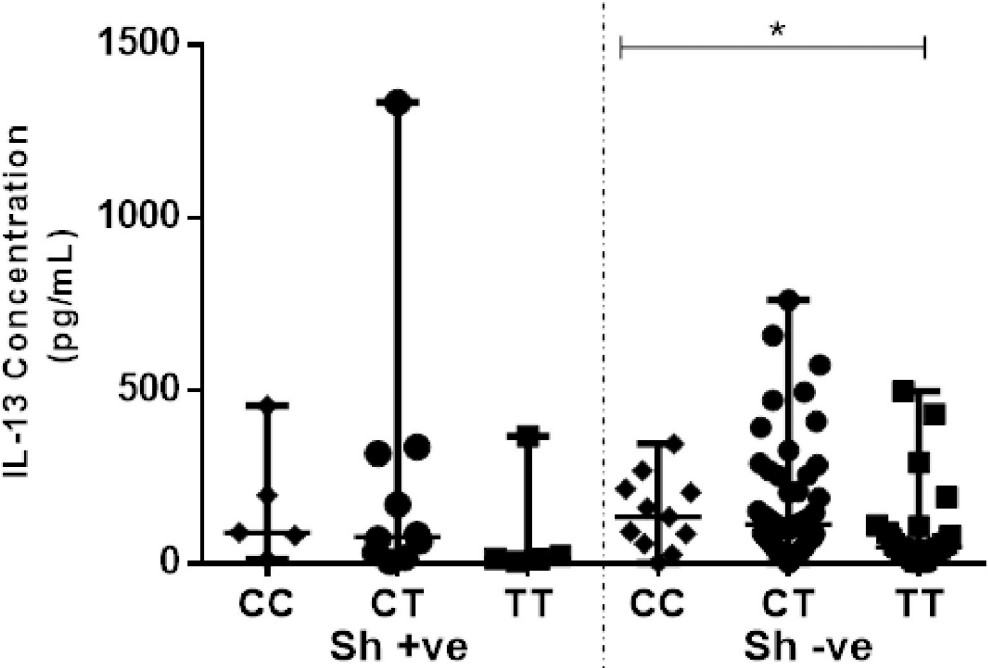

**Fig 4. Comparison of the IL-13 cytokine level in IL-13 -1112C/T variants.**

participants had detectable prostate-specific antigen levels but only 8 (4.1%) participants had prostate-specific antigen levels above 4 ng/mL (S2 Table). Of the 107 participants with detectable IL-13 cytokine levels, 66 participants had detectable prostate-specific antigen levels with 4 participants having prostate-specific antigen levels above 4 ng/mL and 62 participants having prostate-specific antigen levels less than 4 ng/mL. IL-13 cytokine levels were not different between the group with prostate-specific antigen levels < 4 ng/mL 86.32 (29.94–219.80) pg/mL and participants with prostate-specific antigen levels > 4 ng/mL 68.51 (16.89–266.10) pg/mL (p = 0.6927).

Risk of prostate cancer development between participants with prostate-specific antigen levels > 4 ng/mL and < 4ng/mL assessed by IL-13 cytokine concentrations and schistosomiasis status was insignificant $X^2$ (4, n = 66) = 3.996 p = 0.406 (Table 3). The model explained 16% (Nagelkerke $R^2$) of the variance in prostate cancer risk of development and correctly classified 93.9% of the cases. All variables in the model were insignificant (p > 0. 05).

## Discussion

The purpose of this study was to evaluate the frequency of promoter gene polymorphisms, to elucidate promoter gene polymorphisms on IL-13 cytokine levels, to determine susceptibility or protection against schistosomiasis and to determine association to risk of prostate cancer development. Our results illustrated differences in the distribution of IL-13 -1112C/T genotype among study participants and no association of the polymorphism in the schistosomiasis infected and uninfected participants was established. The study involved only adult males who may have developed acquired immunity after repeated exposure [61] and showed low numbers of schistosome infections that may have contributed to the results. However, the findings

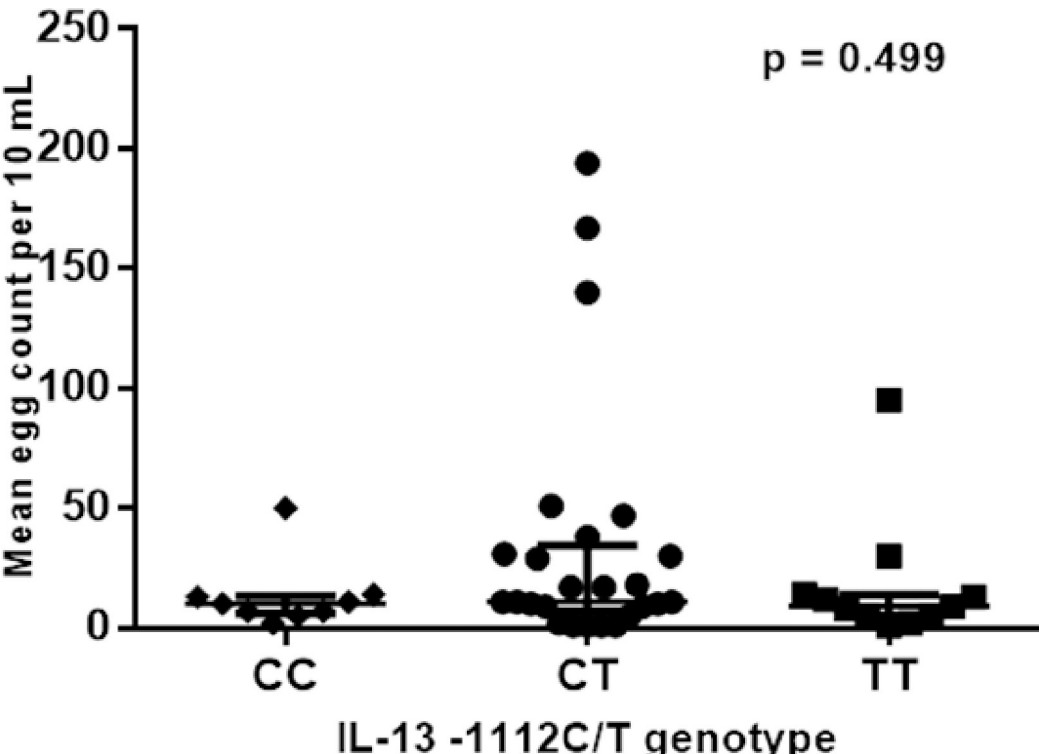

**Fig 5. Comparison of the mean egg burden between IL-13 (rs1800925) variants among *S. haematobium* infected males.**

are consistent with the recent findings by Adedokun *et al.* (2018) who, despite working on different promoter positions found insignificant differences in the IL-13 (rs7719175) genotypic or allelic frequencies between schistosome-infected and uninfected controls; and any association with disease [38]. We found that IL-13 genotype variants are indicative of IL-13 cytokine concentrations, with the highest levels in schistosomiasis uninfected individuals with heterozygous CT variants compared to the homozygous TT variant of the schistosomiasis uninfected group. Additionally, a combination of the homozygote IL-13-1112 CC and heterozygote IL-13-1112CT variants significantly had higher IL-13 cytokine levels compared to IL-13-1112TT variant implying an association between IL-13-1112C/T promoter genotype and IL-13 cytokine levels. The results illustrated that IL13-1112CT genotype exhibited the most frequent distribution among the study population and there were higher schistosomiasis prevalence and infection intensity of individuals with IL13-1112CT variant. Apparently, IL-13-1112C variant

**Table 3. Risk of prostate cancer development (PSA > 4 ng/mL and < 4ng/mL) association to schistosomiasis status, IL-13 cytokine concentration and IL-13 rs1800925 variants.**

| Variable | B | S.E | Wald | df | p. value | Odds Ratio | 95% C.I. |
|---|---|---|---|---|---|---|---|
| Schistosomiasis status | -1.752 | 1.094 | 2.565 | 1 | 0.109 | 0.173 | 0.020–1.480 |
| IL-13 concentrations (pg/mL) | -0.002 | 0.003 | 0.399 | 1 | 0.527 | 0.998 | 0.991–1.004 |
| TT | | | 0.203 | 2 | 0.903 | | |
| CC | -18.528 | 1.283 | 0.000 | 1 | 0.999 | 0.000 | 0.000 |
| CT | .572 | 1.269 | 0.203 | 1 | 0.652 | 1.772 | 0.147–21.336 |
| Constant | -1.505 | 1.185 | 1.612 | 1 | 0.204 | 0.222 | |

individuals were more susceptible to schistosomiasis whereas individuals with IL-13-1112T variant were protected against infection. The findings are consistent with the recent findings of Kouriba *et al.* (2005), who found higher *S. haematobium* infection prevalence among participants with the C/C and C/T genotypes [37]. He *et al.* (2008) also showed that the C/T genotype individuals were more susceptible to *S. haematobium* infection [33]. Similarly, Grant *et al.* in 2012 showed that protection against severe infection with *S. mansoni* was driven by functional IL-13 rs1800925 T polymorphisms [62]. In contrast to our findings, Isnard *et al.* in 2011 showed that IL-13-1112T/T was associated with higher *S. haematobium* infection intensity [34]. Although it was for a different schistosomiasis species, Gaitlin *et al.* showed that IL-13 genotype CT exhibited resistance to *S. mansoni* infection [35]. Our small sample size and perhaps different ethnicity of the study population may explain the different observations.

IL-13 may contribute to disease burden by increasing eosinophil infiltration and promoting fibrosis but could also protect against infection and reduce the risk of schistosomiasis [8,16]. Although there was no statistically significant differences in levels of IL-13 cytokine differences, schistosomiasis infected individuals had higher IL-13 cytokine levels range compared to the uninfected group, thus providing evidence of up-regulated IL-13 levels in schistosome infections to eradicate the parasite tissue lodged *Schistosoma* eggs [7]. In accord to our results, Long *et al.* reported higher IL-13 cytokine levels in *S. japonicum* infected individuals with liver fibrosis compared to those with normal liver tissues [36]. Mutengo *et al.* also showed that high IL-13 levels influenced *S. mansoni* disease progression [17].

We could not establish an association between IL-13-1112C/T variants and susceptibility to schistosomiasis. However, our results showed that schistosomiasis uninfected individuals with IL-13-1112C variant (i.e genotypes CT and CC) had significantly higher IL-13 cytokine levels compared to the TT genotype implying that IL-13–1112 C variants may be protective against *Schistosoma* infections. In contrast to our findings, Kouriba *et al.* reported that protection against schistosomes is increased by the IL13-1055 TT genotype due to the number of less infected individuals with TT genotype [37]. TT genotype individuals had lower IL-13 cytokine levels compared to the CT and CC genotype individuals. We observed that IL-13-1112TT variant had lower mean egg count hence they can control schistosome egg burden compared to CC and CT variants. Our findings contradict those of, Long *et al.* who showed that IL-13 polymorphism rs1800925 T elevates IL-13 production thereby increasing risk of liver fibrosis by *S. japonicum* infected individuals and increasing disease pathology [36]. This is so because IL-13-1112 TT genotype increases transcription of the cytokines leading to elevation of IL-13 cytokine and the cytokine enhances resistance to infection by schistosome in humans [33,63]. Also, He *et al.* [21] and van der Pouw Kraan *et al.* [63] suggested that TT genotype increases transcription of the cytokine leading to elevation of IL-13 cytokine, in turn IL-13 cytokine enhances resistance to infection by schistosome in humans [33,63]. However, lower mean egg count could be because adults do not shed *S. haematobium* eggs well in urine, hence majority of the participants had low infection intensity suggesting that urinary egg counts may not be a good indicator of intensity or disease burden in adults.

Schistosomiasis induced inflammatory cytokines such as macrophage inhibitory cytokine-1 [64], IL-6 [65] and tumour necrosis alpha [66] have been identified as potential mediators between prostatic inflammation and prostate carcinogenesis. More recently, a pro-oncogenic factor for *S. haematobium*, egg secreted infiltrin protein that is an ortholog of interleukin-4-inducing principle of *S. mansoni* egg (IPSE) was shown to initiate bladder urothelial hyperplasia and angiogenesis [67]. Chronic schistosomiasis has been instigated in granuloma formation, tissue eosinophilia, collagen deposition and fibrosis driven by IL-13 cytokine may cause significant morbidity and mortality [68–70]. Additionally, IL-13 promotes and facilitates cancer progression by down regulating immune-surveillance and suppressing cytotoxic T

lymphocytes responses against the tumours [18–20]. An association between increased risk of prostate cancer or aggressive prostate cancer and IL-8 -47CT genotype as well as the IL-10 -1082GG variant was established [71,72]. One of the objectives of this study was to evaluate schistosomiasis up-regulated inflammatory cytokine IL-13 possibly controlled by IL-13 -1112C/T polymorphisms that could directly or indirectly contribute to prostate carcinoma development. Our results showed no significant differences in IL-13 levels among individuals with prostate-specific antigen levels greater and less than 4 ng/mL. Therefore, IL-13 cytokine levels are not associated with prostate-specific antigen levels or inflammation. Logistic regression showed no association between prostate-specific antigen levels greater or less than 4 ng/mL and IL-13 concentrations, IL-13-1112C/T genotype and schistosomiasis status.

Despite different genetic assessment, our results are in accord to Tindall *et al.* (2010) which showed no association between IL-13 alleles and prostate cancer risk [73]. Hence, the IL-13-1112C/T genotype may not be utilised as a biomarker for risk of prostate cancer. To the best of our knowledge this is the first time IL-13 -1112C/T promoter polymorphisms have associated with prostate-specific antigen levels an indicator for risk of prostate cancer. Prostatic *Schistosoma* carcinoma individuals would further elucidate association of the prostate cancer and schistosomiasis. Additionally, inclusion or combination of other inflammatory cytokines elevated due to schistosomiasis and functional polymorphisms may elucidate prostate cancer development due to schistosomiasis. There are some limitations observed in this study caused by the low number of schistosomiasis infected participants and single nucleotide polymorphisms locus may not have provided us with a clear understanding of the genetic effects of IL-13 on its cytokine productions and risk of prostate cancer. As a result, more work is recommended for the IL-13 promoter single nucleotide polymorphisms assessments. Lack of confirmed prostate cancer cases could have limited assessment of the association of IL-13 cytokine levels or IL-13 -1112C/T variants and risk of prostate cancer. Further, only few numbers of individuals were found with prostate-specific antigen above 4 ng/mL that limited our interpretation of the results.

## Conclusion

IL-13 rs1800925/-1112 C variant individuals may have protection from *Schistosoma* infections. There was no association between risk of prostate cancer and IL-13 concentrations and IL-13 rs1800925 genotypes in Zimbabwean male individuals residing in a schistosomiasis endemic area. Therefore, IL-13 levels and IL-13 rs10800925 may not be utilised as a biomarker for risk of prostate cancer in schistosome infections.

## Supporting information

**S1 Table. Determination of IL-13 gene polymorphism adherence to the Hardy-Weinberg Principle.**
(DOCX)

**S2 Table. Prostate-specific antigens detected frequencies by villages.**
(DOCX)

**S1 File. Data.**
(XLSX)

## Acknowledgments

The authors are grateful to the Centre of Immunology and Infection Research group in the Biochemistry Department at the University of Zimbabwe.

## Author Contributions

**Conceptualization:** Emilia T. Choto, Takafira Mduluza, Moses J. Chimbari.

**Data curation:** Emilia T. Choto, Takafira Mduluza.

**Formal analysis:** Emilia T. Choto.

**Funding acquisition:** Moses J. Chimbari.

**Methodology:** Emilia T. Choto.

**Supervision:** Takafira Mduluza, Moses J. Chimbari.

**Writing – original draft:** Emilia T. Choto.

**Writing – review & editing:** Takafira Mduluza, Moses J. Chimbari.

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
