## [Decision Letter · Decision Letter 0]

8 Feb 2021

PONE-D-20-36360

Interleukin-13 rs1800925/-1112C/T promoter single nucleotide polymorphism variant linked to anti-schistosomiasis adult males in Murehwa District, Zimbabwe.

PLOS ONE

Dear Ms Emilia Choto,

Thank you for submitting your manuscript to PLOS ONE. After careful consideration, we feel that it has merit but does not fully meet PLOS ONE’s publication criteria as it currently stands. Therefore, we invite you to submit a revised version of the manuscript that addresses the points raised during the review process.

Specifically, questions about the methodology and study design should be addressed; explanations should be given for why numbers don't tally in your manuscript. You enrolled  366 participants but report SNP results for 356 participants (Page 12: Lines 284-285) and no reason(s) was/were given for the remaining one that was missing; probably due to unsuccessful genotyping. Of the 366 participants that were enrolled, IL-13 cytokine concentrations were reported for only 107 participants and no reason(s) was/were given for the remaining unreported results. In addition, the authors did not state how many participants out of the 107 had schistosomiasis and how many were uninfected.It was observed that certain data presented in this study had earlier been published by the authors [Choto et al. Infectious Agents and Cancer (2020) 15:59; PMID: 33042215], but no mention was made about this in the Results section (Page 12: Lines 281-282) so as to avoid any semblance of duplicate publishingYou need to also clarify the connection between the stated aims of your study; not much attention was given to the prostate antigen aspect. The study looks like two different studies (one on the association between IL-13rs1800925/-1112C/T promoter SNPs and Schistosoma haematobium infection, and the other on association between IL-13rs1800925 polymorphism and prostate-specific antigen levels) but the nexus between the two does not come out clearly in the literature review and discussion.There are also grammatical errors that need to be edited.

We look forward to receiving your revised manuscript.

Kind regards,

Chiaka Ijeoma Anumudu

Academic Editor

PLOS ONE

Journal Requirements:

2. Please provide additional details regarding participant consent. In the ethics statement in the Methods and online submission information, please ensure that you have specified whether consent was informed.

3. In the Methods, please clarify:

- Why written consent could not be obtained

- Whether the Institutional Review Board (IRB) approved use of verbal consent

- How verbal consent was documented

For more information, please see our guidelines for human subjects research: https://journals.plos.org/plosone/s/submission-guidelines#loc-human-subjects-research

4.In your Methods section, please provide additional information about the participant recruitment method and the demographic details of your participants. Please ensure you have provided sufficient details to replicate the analyses such as: a) the recruitment date range (month and year),b) a table of relevant demographic details, c) a statement as to whether your sample can be considered representative of a larger population, d) a description of how participants were recruited, and e) descriptions of where participants were recruited and where the research took place.

5. Please provide a sample size and power calculation in the Methods, or discuss the reasons for not performing one before study initiation.

6. Please ensure you have discussed any potential limitations of your study in the Discussion, including study design, sample size and/or potential confounders.

7.We note that you have indicated that data from this study are available upon request. PLOS only allows data to be available upon request if there are legal or ethical restrictions on sharing data publicly. For more information on unacceptable data access restrictions, please see http://journals.plos.org/plosone/s/data-availability#loc-unacceptable-data-access-restrictions.

8. We note you have included a table to which you do not refer in the text of your manuscript. Please ensure that you refer to Table 4 in your text; if accepted, production will need this reference to link the reader to the Table.

9. Please include a copy of Table 2 which you refer to in your text on page 15.

Reviewers' comments:

Reviewer's Responses to Questions

**Comments to the Author**

1. Is the manuscript technically sound, and do the data support the conclusions?

Reviewer #1: Yes

Reviewer #2: Yes

Reviewer #3: Partly

2. Has the statistical analysis been performed appropriately and rigorously? 

Reviewer #1: Yes

Reviewer #2: Yes

Reviewer #3: Yes

3. Have the authors made all data underlying the findings in their manuscript fully available?

Reviewer #1: Yes

Reviewer #2: Yes

Reviewer #3: Yes

4. Is the manuscript presented in an intelligible fashion and written in standard English?

Reviewer #1: Yes

Reviewer #2: Yes

Reviewer #3: Yes

5. Review Comments to the Author

Reviewer #1: The study attempted to evaluate the IL-13 in susceptibility or resistance against schistosomiasis and association to risk of prostate cancer development. The study should provide promising information on understanding the role of the cytokine in host response to the disease schistosomiasis.

Introduction

Line 99-100 "Host genetic variability of encoding genes such as polymorphisms located in the promoter region modify gene transcription and cytokine production in parasitic and autoimmune diseases" is not explicit and appeared inconclusive, please rephrase

Line 108-110 add reference to the statement "Single nucleotide polymorphisms (SNPs) are the most common genetic variations..... sequences and protein structure (ref.)

Line 116- Isnard et al (2011)... and not Isnard et al., in 2011

Line 144- Remove schistosome from the statement "S. haematobium schistosome egg burden was associated"

Materials and Methods

Line 164- Inclusion criteria: Reason on focusing on adults aged 18 and above should be clearly stated

Line 174-175: Schistosome infected individuals were treated with praziquantel(PZQ) at the standard single oral dose of 40mg/kg per body weigh (ref.)

Line 162-163: Detailed description of study areas was not well provided in this study. Providing map, daily activities of the study population will help

Line 360-362: "Binomial logistic regression was used to predict the risk of prostate cancer development between participants with prostate specific antigen levels > 4 ng/mL and < 4ng/mL using IL362 13 cytokine concentrations and schistosomiasis status as the predictor value". The statement is more of methodology than results, rephrase.

Discussion: The discussion is well written but more recent literatures on study should be included to help support the outcome of the present study.

Reviewer #2: The manuscript is technically sound and the data support the conclusion. However, the study looks like two different studies (one on the association between IL-13rs1800925/-1112C/T promoter SNPs and Schistosoma haematobium infection, and the other on association between IL-13rs1800925 polymorphism and prostate-specific antigen levels) but the nexus between the two does not come out clearly in the literature review and discussion. The author should also consider and mention that adult humans do not shed S. haematobium eggs well in urine and urinary egg counts may therefore not be a good indicator of infection intensity or disease burden in adults. In addition it is noteworthy that although IL-13 is an important player in schistosomiasis carcinogenesis, there are other important players such as infiltrin which should have been mentioned in discussion. Being a cross-sectional study, no influence could have been detected and therefore the word "influence" in lines 34 and 152 should be changed to "association".

The analyses were performed appropriately and and rigorously. however, the power of the study could have been compromised by the small number of infected individuals (n=50). The data underlying the findings were made available but it would have been better if the age distribution of the subjects was also presented.

The manuscript is presented in an intelligible fashion and in standard English. However, there are several grammatical errors and omissions that should be corrected. For example, the term "schistosomiasis infected" should be changed to "S. haematobium-infected"

Reviewer #3: ABSTRACT

Page 2: Lines 32-34

The authors should correct "Our study evaluated the frequency of the IL-13 rs1800925/-1112 C/ T promoter single nucleotide polymorphisms (SNPs) and assessed the influence of the variants on IL-13 cytokine levels." to read "Our study evaluated the frequency of the IL-13 rs1800925/-1112 C/ T promoter single nucleotide polymorphisms (SNPs) among schistosomiasis infected individuals and assessed the influence of the variants on IL-13 cytokine levels."

Page 2: Lines 46-48

The authors should indicated whether the sentence "There were significantly (p<0.05) higher IL-13 cytokine levels among participants with the genotypes CC and CT; median 92.25 pg/mL and 106.5 pg/mL, respectively, compared to TT variant individuals; 44.78 pg/mL." is with respect to schistosomiasis infected individuals or uninfected.

INTRODUCTION

Page 4: Line 82

Please restructure "...Schistosomiasis is a neglected parasitic tropical disease" to read "...Schistosomiasis is a neglected tropical parasitic disease"

Page 4: Line 91

Please restructure "...Schistosomiasis acute infections" to read "...Acute schistosomiasis infections"

Page 4: Lines 98-101

Needs grammatical restructuring.

Page 5: Line 109

Please restructure "...located in the coding regions of the genes" to read "...located in the coding regions of genes"

Page 5: Lines 122-125

The sentence "In contrast, to the above findings recently Adedokun et al. in 2018 found no statistical difference in the IL-13 rs7719175 genotypic or allelic frequencies between schistosome-infected and uninfected controls or any association with disease [26]." would be better in the Discussion Section.

Page 5: Lines 125-127

The sentence "Furthermore, no association between IL-13 rs1800925 C/T and autoimmune diseases such as Graves’ disease risk [27] and rheumatoid arthritis [18] has been established." should be removed because it does not appear to have any relevance there.

Page 6: Lines 150 and 151

The sentence "Therefore, the aim of the study were..." should be corrected to read "Therefore, the aim of the study was...".

MATERIALS AND METHODS

Page 7: Lines 161

The authors should also give a description of the enrolment or case definition of participants with prostate cancer.

RESULTS

Page 12: Lines 281-282

It was observed that certain data presented in this study had earlier been published by the authors [Choto et al. Infectious Agents and Cancer (2020) 15:59; PMID: 33042215]. The authors should have made mention of this in this section.

Page 12: Lines 285-285

The authors had earlier indicated that 366 participants were enrolled but genotypic results for IL-13 -1112C/T polymorphism were given for only 356 participants. The authors should give reasons for this discrepancy.

Page 13: Line 308

The authors presented IL-13 cytokine concentrations results for only 107 participants out of the 366 participants that were enrolled. The authors should also give reasons for this discrepancy.

Page 15: Line 367

The authors should review this Table to ensure that the Title is a true reflection of the contents.

6. PLOS authors have the option to publish the peer review history of their article (what does this mean?). If published, this will include your full peer review and any attached files.

Reviewer #1: No

Reviewer #2: No

Reviewer #3: **Yes: **Dr. Segun Isaac OYEDEJI

---

## [Author Response · Author response to Decision Letter 0]

23 Mar 2021

We appreciate the time and effort the reviewers and the editor have dedicated to providing feedback on ways to improve our paper. We have incorporated changes that reflect the detailed suggestions and comments you have provided as indicated below:

Reviewer #1 Comments: 

The study attempted to evaluate the IL-13 in susceptibility or resistance against schistosomiasis and association to risk of prostate cancer development. The study should provide promising information on understanding the role of the cytokine in host response to the disease schistosomiasis.

Response: We are appreciative for the acknowledgement on the topic we are reporting on.

Introduction

Line 99-100 "Host genetic variability of encoding genes such as polymorphisms located in the promoter region modify gene transcription and cytokine production in parasitic and autoimmune diseases" is not explicit and appeared inconclusive, please rephrase.

Response: We have revised the sentence to read as follows: Host genetic variability such as single nucleotide polymorphisms located in the promoter region of encoding genes may modify gene transcription and cytokine production in parasitic infections and autoimmune diseases. 

Line 108-110 add reference to the statement "Single nucleotide polymorphisms (SNPs) are the most common genetic variations..... sequences and protein structure (ref.)

Response: We have included the reference: 31. Howard TD, Whittaker PA, Zaiman AL, Koppelman GH, Xu J, Hanley MT et al. Identification and association of polymorphisms in the interleukin-13 gene with asthma and atopy in a Dutch population. Am J Respir Cell Mol Biol. 2001;25(3):377–84. PMID: 11588017 doi: 10.1165/ajrcmb.25.3.4483.

Line 116- Isnard et al (2011)... and not Isnard et al., in 2011

Response: The reference has been fixed.

Line 144- Remove schistosome from the statement "S. haematobium schistosome egg burden was associated"

Response: Schistosome has been removed from the statement.

Materials and Methods

Line 164- Inclusion criteria: Reason on focusing on adults aged 18 and above should be clearly stated.

Response: The reason on the for focusing on adults aged 18 years and above is because the study is a sub-study of prostate cancer and schistosomiasis cross-sectional study that included only male adult participants aged 18 years and above. Adults aged 18 years and above were included into the study because they were able to understand the study and voluntarily give informed consent.

Line 174-175: Schistosome infected individuals were treated with praziquantel(PZQ) at the standard single oral dose of 40mg/kg per body weight (ref.)

Response: Reference has been provided: 58. Crompton DWT, World Health Organization. Preventative chemotherapy in human helminthiasis: coordinated use of anthelminthic drugs in control interventions: a manual for health professionals and programme managers. World Health Organization 2006. Available at: https://apps.who.int/iris/handle/10665/43545.

Line 162-163: Detailed description of study areas was not well provided in this study. Providing map, daily activities of the study population will help

Response: Even though we have not provided a map we have added information on the study population and area: The study area was chosen because it has a high schistosomiasis burden of 47.4 % in school aged children a representative of the study population [52]. Overall population of Murehwa District is 199 607 and about 94 000 individuals are men [53]. The District consists of more than 90 % of the area is rural where majority of the residents rely on nearby rivers for their domestic needs and farming activities. Sanitation in the area is poor with the majority of residents relying on unsafe drinking water and open defecation. The main study assessed possibility of prostate cancer development due to schistosome infections hence, only male adult individuals were enrolled

Line 360-362: "Binomial logistic regression was used to predict the risk of prostate cancer development between participants with prostate specific antigen levels > 4 ng/mL and < 4ng/mL using IL362 13 cytokine concentrations and schistosomiasis status as the predictor value". The statement is more of methodology than results, rephrase.

Response: Agreed and statement has been rephrased to read as follows: ‘Risk of prostate cancer development between participants with prostate specific antigen levels > 4 ng/mL and < 4ng/mL assessed by IL-13 cytokine concentrations and schistosomiasis status was insignificant.’

Discussion: 

The discussion is well written but more recent literatures on study should be included to help support the outcome of the present study. 

Response: We appreciate the acknowledgement and we included the limited available information in the discussion on the following information: account for low number of schistosomiasis infected participants, low infection intensities, information on other inflammatory mediators between prostatic inflammation and prostate carcinogenesis and the study limitations.

Reviewer # 2 Comments:

The manuscript is technically sound and the data support the conclusion. However, the study looks like two different studies (one on the association between IL-13rs1800925/-1112C/T promoter SNPs and Schistosoma haematobium infection, and the other on association between IL-13rs1800925 polymorphism and prostate-specific antigen levels) but the nexus between the two does not come out clearly in the literature review and discussion. 

Response: Thank you and we agree on the assessment. We have now included additional information to demonstrate the nexus between the IL-13 SNPs or concentration, schistosomiasis and prostate specific antigen. Please note that the association of IL-13 SNPs, concentrations, schistosomiasis and prostate specific antigen has very limited literature hence, we believe that our manuscript will build literature on the subject. ‘Screening for prostate cancer is initially done by using prostate-specific antigen levels to detect the diseases early stage for better management and reduction of disease specific mortality. Whilst prostate specific-antigen levels above 4.0 ng/mL serves as a reference point for further prostate management. To further determine risk of prostate cancer, combination of prostate specific antigen with single nucleotide polymorphisms was shown to be effective in men with prostate specific antigen levels greater than 4 ng/mL. Combination of genetics and the PSA test is useful for predicting the risk of prostate cancer that enables stratifying the population into different risk groups that may be a basis for the development of personalized screening for prostate cancer.

The author should also consider and mention that adult humans do not shed S. haematobium eggs well in urine and urinary egg counts may therefore not be a good indicator of infection intensity or disease burden in adults. In addition it is noteworthy that although IL-13 is an important player in schistosomiasis carcinogenesis, there are other important players such as infiltrin which should have been mentioned in discussion. 

Response: Thank you very much, we appreciate the input and have included the information in the manuscript such as the macrophage inhibitory cytokine-1, IL-6, tumour necrosis alpha, S. haematobium, egg secreted infiltrin protein, IL-10 and IL-8. 

Being a cross-sectional study, no influence could have been detected and therefore the word "influence" in lines 34 and 152 should be changed to "association".

Response: The word influence has been changed to association. 

The analyses were performed appropriately and rigorously however, the power of the study could have been compromised by the small number of infected individuals (n=50). The data underlying the findings were made available but it would have been better if the age distribution of the subjects was also presented.

Response: We agree with the reviewer and we have now included a table (Table 1) with the age distribution of the study participants.

The manuscript is presented in an intelligible fashion and in standard English. However, there are several grammatical errors and omissions that should be corrected. For example, the term "schistosomiasis infected" should be changed to "S. haematobium-infected".

Response: All authors have proofread the manuscript to correct grammatical errors.

Reviewer # 3 Comments:

ABSTRACT

Page 2: Lines 32-34

The authors should correct "Our study evaluated the frequency of the IL-13 rs1800925/-1112 C/ T promoter single nucleotide polymorphisms (SNPs) and assessed the influence of the variants on IL-13 cytokine levels." to read "Our study evaluated the frequency of the IL-13 rs1800925/-1112 C/ T promoter single nucleotide polymorphisms (SNPs) among schistosomiasis infected individuals and assessed the influence of the variants on IL-13 cytokine levels."

Response: Statement has been corrected to read :‘Our study evaluated the frequency of the IL-13 rs1800925/-1112 C/ T promoter single nucleotide polymorphisms (SNPs) among schistosomiasis infected individuals and assessed the association of the variants on IL-13 cytokine levels’.

Page 2: Lines 46-48

The authors should indicated whether the sentence "There were significantly (p<0.05) higher IL-13 cytokine levels among participants with the genotypes CC and CT; median 92.25 pg/mL and 106.5 pg/mL, respectively, compared to TT variant individuals; 44.78 pg/mL." is with respect to schistosomiasis infected individuals or uninfected.

Response: Statement has been specified to read as follows: There were significantly (p<0.05) higher IL-13 cytokine levels among both infected and uninfected participants with the genotypes CC and CT; median 92.25 pg/mL and 106.5 pg/mL, respectively, compared to TT variant individuals; 44.78 pg/mL.

INTRODUCTION

Page 4: Line 82

Please restructure "...Schistosomiasis is a neglected parasitic tropical disease" to read "...Schistosomiasis is a neglected tropical parasitic disease"

Response: Statement has been corrected

Page 4: Line 91

Please restructure "...Schistosomiasis acute infections" to read "...Acute schistosomiasis infections"

Response: Statement has been restructured. 

Page 4: Lines 98-101

Needs grammatical restructuring.

Response: Statement has been restricted to read as follows: Schistosome infection associated cytokines are connected to cytokine gene polymorphisms of individual variability that influence immune responses and disease outcomes [11]. Cytokine gene polymorphisms such as single nucleotide polymorphisms located in the promoter region of encoding genes may modify gene transcription and cytokine production in parasitic infections and autoimmune diseases. 

Page 5: Line 109

Please restructure "...located in the coding regions of the genes" to read "...located in the coding regions of genes"

Response: Statement has been restructured accordingly. 

The sentence "In contrast, to the above findings recently Adedokun et al. in 2018 found no statistical difference in the IL-13 rs7719175 genotypic or allelic frequencies between schistosome-infected and uninfected controls or any association with disease [26]." would be better in the Discussion Section.

Response: We believe the sentence adds value into the introduction by showing differences in the results conducted by other studies hence also validating the need to conduct our study. However, we have included the results of the study to compare to our findings in the discussion section. 

The sentence "Furthermore, no association between IL-13 rs1800925 C/T and autoimmune diseases such as Graves’ disease risk [27] and rheumatoid arthritis [18] has been established." should be removed because it does not appear to have any relevance there.

Response: We agree with the reviewer and the statement has been removed.

The sentence "Therefore, the aim of the study were..." should be corrected to read "Therefore, the aim of the study was...".

Response: Sentence has been corrected ‘were’ had been replaced with ‘was’

MATERIALS AND METHODS

Page 7: Lines 161

The authors should also give a description of the enrolment or case definition of participants with prostate cancer.

Response: We have included information on prostate cancer. ‘Screening for prostate cancer is initially done by using prostate-specific antigen levels to detect the diseases early stage for better management and reduction of disease specific mortality [48]. Whilst prostate specific-antigen levels above 4.0 ng/mL serves as a reference point for further prostate management [49]. To further determine risk of prostate cancer, combination of prostate specific antigen with single nucleotide’.

RESULTS

Page 12: Lines 281-282

It was observed that certain data presented in this study had earlier been published by the authors [Choto et al. Infectious Agents and Cancer (2020) 15:59; PMID: 33042215]. The authors should have made mention of this in this section.

Response: Statements have been included in the manuscript to acknowledge the previously presented data by Choto et al. (2020).

Page 12: Lines 285-285

The authors had earlier indicated that 366 participants were enrolled but genotypic results for IL-13 -1112C/T polymorphism were given for only 356 participants. The authors should give reasons for this discrepancy.

Response: Noted and we have included a reason for the discrepancy in the results and participants enrolled in the manuscript. ‘There was unsuccessful genotyping of 10 samples due to unsuccessful genotyping.

Page 13: Line 308

The authors presented IL-13 cytokine concentrations results for only 107 participants out of the 366 participants that were enrolled. The authors should also give reasons for this discrepancy.

Response: Two hundred and fifty-nine (259) samples had undetectable IL-13 concentrations hence the IL-13 results are for only 107 samples. 

Page 15: Line 367

The authors should review this Table to ensure that the Title is a true reflection of the contents. 

Response: The table title has been revised and it now reads ‘Table 3: Risk of prostate cancer development (PSA > 4 ng/mL and < 4ng/mL) association to schistosomiasis status, IL-13 cytokine concentration and IL-13 rs1800925 variants’

Academic Editor Comments and additional requirements

1. Please ensure that your manuscript meets PLOS ONE's style requirements, including those for file naming. The PLOS ONE style templates can be found at:

Response: We have tried our best to make sure that the manuscript is compliant with the PLOS ONE guidelines.

2. Please provide additional details regarding participant consent. In the ethics statement in the Methods and online submission information, please ensure that you have specified whether consent was informed.

Response: We have included the following statement in the ethics section: Prior to enrolment, the study aims and procedures were explained to all participants in the local language (Shona). Written informed consent was obtained from participants.

3. In the Methods, please clarify:

- Why written consent could not be obtained

- Whether the Institutional Review Board (IRB) approved use of verbal consent

- How verbal consent was documented

For more information, please see our guidelines for human subjects research: https://journals.plos.org/plosone/s/submission-guidelines#loc-human-subjects-research

 Response: Written consent was obtained.

4. In your Methods section, please provide additional information about

i)the participant recruitment method and

Response: Participant recruitment information has been provided the methodology section. Participants from rural villages in Murehwa Disctrict were invited to be part of the study through invitation by the village health workers to report to 8 sampling centres that are used by the community for meetings or health education and immunisation programs (such centres include schools or primary health centres) for enrolment into the study. The sampling centres were Jekwa rural clinic, Dombwe rural clinic, Mutize primary school, Kareza gathering point, Kapasura gathering point, Magaya primary school, Guzha primary school and Inyagui primary school. Participants enrolled into the study provided their age or date of birth, samples for parasitological diagnosis and blood sample for serological assays. 

ii)the demographic details of your participants.

Response: Demographic details of the participants have been provided in the results section (Table 1).

Please ensure you have provided sufficient details to replicate the analyses such as: 

a) the recruitment date range (month and year),

Response: Recruitment month and year were included: November 2019

b) a table of relevant demographic details,

Response: Table of demographics details has been included (Table 1).

c)a statement as to whether your sample can be considered representative of a larger population, 

Response: The sample can be considered representative of a larger population in the study area. The statement is included in the methodology section. 

d) a description of how participants were recruited, and 

Response: We have now provided the information in the methodology section and in the above response to query # 4.

e) descriptions of where participants were recruited and where the research took place.

 Response: We have now provided the information in the methodology section and in the above response to query # 4.

5. Please provide a sample size and power calculation in the Methods,or discuss the reasons for not performing one before study initiation.

Response: Sample size calculations have been provided in the methodology section. ‘A sample size of 245 including 20 % added on to account for drop out was calculated [54] based on the inferred schistosomiasis prevalence for adults of 15.8 % (a third of the 47.4 % for the school aged children)’

6. Please ensure you have discussed any potential limitations of your study in the Discussion, including study design, sample size and/or potential confounders.

 Response: Limitations have been included in the discussion. 

‘There are some limitations observed in this study caused by the low number of schistosomiasis infected participants and single nucleotide polymorphisms locus may not have provided us with a clear understanding of the genetic effects of IL-13 on its cytokine productions and risk of prostate cancer. As a result, more work is recommended for the IL-13 promoter single nucleotide polymorphisms assessments. Lack of confirmed prostate cancer cases could have limited assessment of the association of IL-13 cytokine levels or IL-13 -1112C/T variants and risk of prostate cancer. Further, only few numbers of individuals were found with prostate specific antigen above 4 ng/mL that limited our interpretation of the results’

7.We note that you have indicated that data from this study are available upon request. PLOS only allows data to be available upon request if there are legal or ethical restrictions on sharing data publicly. For more information on unacceptable data access restrictions, please see http://journals.plos.org/plosone/s/data-availability#loc-unacceptable-data-access-restrictions.

Response: Data has been provided as supporting files (S1 File)

• Specifically, questions about the methodology and study design should be addressed; explanations should be given for why numbers don't tally in your manuscript. 

o You enrolled 366 participants but report SNP results for 356 participants (Page 12: Lines 284-285) and no reason(s) was/were given for the remaining one that was missing; probably due to unsuccessful genotyping.

Response: We have addressed the issue in the above response to comments raised by reviewer # 3

o Of the 366 participants that were enrolled, IL-13 cytokine concentrations were reported for only 107 participants and no reason(s) was/were given for the remaining unreported results. 

Response: We have addressed the issue in the above response to comments raised by reviewer # 3

o In addition, the authors did not state how many participants out of the 107 had schistosomiasis and how many were uninfected.

Response: We have now provided the information in the manuscript. Participants with schistosomiasis (n = 20) had lower but not significant levels of IL-13, 75.64 (14.52 - 287.50) pg/mL compared to schistosomiasis uninfected group (n = 87) 89.88 (40.03 - 206.20) pg/mL; p = 0.481.

• It was observed that certain data presented in this study had earlier been published by the authors [Choto et al. Infectious Agents and Cancer (2020) 15:59; PMID: 33042215], but no mention was made about this in the Results section (Page 12: Lines 281-282) so as to avoid any semblance of duplicate publishing.

Response: We have acknowledged the results that were earlier published by Choto et. al (2020). Also, we have included reference in the results section.

• You need to also clarify the connection between the stated aims of your study; not much attention was given to the prostate antigen aspect. The study looks like two different studies (one on the association between IL-13rs1800925/-1112C/T promoter SNPs and Schistosoma haematobium infection, and the other on association between IL-13rs1800925 polymorphism and prostate-specific antigen levels) but the nexus between the two does not come out clearly in the literature review and discussion.

Response: We have addressed the query by providing information on the nexus between polymorphisms and prostate specific antigen levels in the above response to comments raised by reviewer # 2 above

• There are also grammatical errors that need to be edited.

Response: We have tried our best to revise the manuscript for grammatical errors.

---

## [Decision Letter · Decision Letter 1]

12 May 2021

Interleukin-13 rs1800925/-1112C/T promoter single nucleotide polymorphism variant linked to anti-schistosomiasis in adult males in Murehwa District, Zimbabwe.

PONE-D-20-36360R1

Dear Ms Choto,

We’re pleased to inform you that your manuscript has been judged scientifically suitable for publication and will be formally accepted for publication once it meets all outstanding technical requirements.

Kind regards,

Chiaka Ijeoma Anumudu

Academic Editor

PLOS ONE

Additional Editor Comments (optional):

Reviewers' comments:

Reviewer's Responses to Questions

**Comments to the Author**

1. If the authors have adequately addressed your comments raised in a previous round of review and you feel that this manuscript is now acceptable for publication, you may indicate that here to bypass the “Comments to the Author” section, enter your conflict of interest statement in the “Confidential to Editor” section, and submit your "Accept" recommendation.

Reviewer #3: All comments have been addressed

2. Is the manuscript technically sound, and do the data support the conclusions?

Reviewer #3: Yes

3. Has the statistical analysis been performed appropriately and rigorously? 

Reviewer #3: Yes

4. Have the authors made all data underlying the findings in their manuscript fully available?

Reviewer #3: Yes

5. Is the manuscript presented in an intelligible fashion and written in standard English?

Reviewer #3: Yes

6. Review Comments to the Author

Reviewer #3: The initial comments have been addressed but there are few, though minor observations that needs to be addressed. The authors should please make the following minor corrections:

INTRODUCTION

Page 5: Line 128

Please insert a full stop (.) after et al

Page 6: Lines 129-130

Please remove the comma (,) after In contrast and place it after findings in line 130.

MATERIALS AND METHODS

Page 7: Lines 161 and 170

I think that it is better to start this section with Study design, study area and study population (as it was in the original

submission), rather than starting with Ethical approval.

Page 7: Lines 176-177

Please reconstruct the sentence "The District consists of more than 90 % of the area is..."

Page 8: Line 186

Please change "give" to "gave"

RESULTS

Page 13: Line 323

Please insert frequency after genotypic

Page 14: Line 331 (Table 1)

Please remove the semicolon in the parenthesis under Total.

DISCUSSION

Pages 17 and 18: Lines 403-404.

Please reconstruct the sentence "... to elucidate promoter gene polymorphisms on IL-13 cytokine levels, role to

susceptibility or protective against schistosomiasis..."

Page 21 Line 486.

Please insert "which" after (2010)

7. PLOS authors have the option to publish the peer review history of their article (what does this mean?). If published, this will include your full peer review and any attached files.

Reviewer #3: **Yes: **Segun Isaac OYEDEJI

---

## [Editor Report · Acceptance letter]

20 May 2021

PONE-D-20-36360R1 

Interleukin-13 rs1800925/-1112C/T promoter single nucleotide polymorphism variant linked to anti-schistosomiasis in adult males in Murehwa District, Zimbabwe. 

Dear Dr. Choto:

I'm pleased to inform you that your manuscript has been deemed suitable for publication in PLOS ONE. Congratulations! Your manuscript is now with our production department. 

Kind regards, 

on behalf of

Dr. Chiaka Ijeoma Anumudu 

Academic Editor

PLOS ONE